# Twitter Use in Hurricane Isaac and Its Implications for Disaster Resilience

**Kejin Wang** [1,*] , **Nina S. N. Lam** [1] , **Lei Zou** [2] **and Volodymyr Mihunov** [1]

[1] Department of Environmental Sciences, Louisiana State University, Baton Rouge, LA 70803, USA; nlam@lsu.edu (N.S.N.L.); vmihun1@lsu.edu (V.M.)

[2] Department of Geography, Texas A&M University, College Station, TX 77843, USA; lzou@tamu.edu

\* Correspondence: kwang27@lsu.edu

**Abstract:** Disaster resilience is the capacity of a community to "bounce back" from disastrous events. Most studies rely on traditional data such as census data to study community resilience. With increasing use of social media, new data sources such as Twitter could be utilized to monitor human response during different phases of disasters to better understand resilience. An important research question is: Does Twitter use correlate with disaster resilience? Specifically, will communities with more disaster-related Twitter uses be more resilient to disasters, presumably because they have better situational awareness? The underlying issue is that if there are social and geographical disparities in Twitter use, how will such disparities affect communities' resilience to disasters? This study examines the relationship between Twitter use and community resilience during Hurricane Isaac, which hit Louisiana and Mississippi in August 2012. First, we applied the resilience inference measurement (RIM) model to calculate the resilience indices of 146 affected counties. Second, we analyzed Twitter use and their sentiment patterns through the three phases of Hurricane Isaac— preparedness, response, and recovery. Third, we correlated Twitter use density and sentiment scores with the resilience scores and major social–environmental variables to test whether significant geographical and social disparities in Twitter use existed through the three phases of disaster management. Significant positive correlations were found between Twitter use density and resilience indicators, confirming that communities with higher resilience capacity, which are characterized by better social–environmental conditions, tend to have higher Twitter use. These results imply that Twitter use during disasters could be improved to increase the resilience of affected communities. On the other hand, no significant correlations were found between sentiment scores and resilience indicators, suggesting that further research on sentiment analysis may be needed.

**Keywords:** disaster resilience; Hurricane Isaac; social media; Twitter use; sentiment; resilience inference measurement (RIM)

## 1. Introduction

The notion of resilience was introduced by Holling [1] in the field of ecology as the ability of a system to absorb change while maintaining its integrity. Since then, it has been discussed and utilized widely in many other disciplines, including psychology and psychiatry [2,3], geography [4,5], and sociology [6], just to name a few. Although the concept of resilience varies among different disciplines, we adopt the U.S. Federal Emergency Management Agency's (FEMA) definition of community resilience as "the ability of a community to 'bounce back' from disaster impacts, by affecting resilience-building activities through the four phases of emergency management—preparedness, response, recovery, and mitigation" [7,8].

Studies on disaster resilience are abundant. However, most of them have been relying on traditional social and economic data to estimate the resilience capacity of communities [5,9–13]. A major shortcoming of the traditional approach is that data describing real-time human awareness and responses during disaster events are not available through

traditional databases or collection methods. Disaster awareness and responses are critical for emergency management because people with higher awareness and active responses are more likely to take actions to mitigate disaster impacts [14]. Social media such as Twitter provides a new data source for studying real-time disaster assessment, response, and recovery [15–19].

However, social-media data are big and noisy; they need to be mined and integrated with traditional data for studying disaster resilience. Mining social-media data, including data cleaning, text analysis, and data visualization, is challenging due to their huge quantity, unbalanced user composition, redundancy, presence of incorrect or imprecise user locations, and false information [18–20]. There is a great need to develop efficient data-mining algorithms and frameworks to extract useful information from social-media data for resilience studies. More importantly, a major question regarding social media use for resilience analysis is: Are there distinct social and geographical disparities in disaster-related social media use, and if so, how will the disparate social media use affect the resilience of communities?

This paper addresses the research question by examining the correlation between Twitter use pattern and resilience level by county during Hurricane Isaac, which occurred in August 2012 and affected Louisiana and nearby states along the Gulf of Mexico. We hypothesize that counties with higher resilience levels are associated with higher Twitter use densities and more positive sentiment scores. The study applies the resilience inference measurement (RIM) method to calculate county-level resilience scores in the affected area (146 counties). Twitter data during Hurricane Isaac were purchased, and key variables including Twitter use density and sentiment scores were extracted and analyzed across the three phases of emergency management—preparation, response, and recovery. Finally, Pearson's correlation analysis between resilience variables and Twitter variables was conducted to test the hypothesis. The study is among the first to analyze the relationship between resilience score and Twitter use pattern. Findings from the study will shed light on the issue of social and geographical disparities of social media use and add insights into ways to improve the use of Twitter for enhancing resilience to disasters.

## 2. Background

### 2.1. Disaster-Resilience Measurement

Enhancing resilience to natural or man-made hazards is an important societal goal that attracts many scholars' interests. When disaster strikes, people in different regions sustain damages and recover unevenly because of their differences in resilience capacity [5,9,10,21–23]. To improve the resilience capacity of communities, one must identify key resilience factors and evaluate their baselines. Numerous frameworks and models have been developed to measure the resilience of communities to various hazards. Cai et al. [10] classify resilience estimation models into two categories: non-inferential versus inferential models. Non-inferential models for resilience measurement do not require statistical assumptions; they are often generated by weighting several variables and then combining them into a composite index. The derived indices from non-inferential models are applicable only to the study area chosen for the study; the results cannot be inferred to other study areas [5,24]. Examples are the Baseline Resilience Indicators for Communities (BRIC) [11], the National Oceanic and Atmospheric Administration (NOAA)'s Coastal Resilience Index (CRI), and the Resilience Capacity Index [2].

Inferential statistical models derive resilience indices using inferential statistics such as regression modeling and discriminant analysis. For instance, the resilience inference measurement (RIM) model evaluates the resilience level of a community based on a set of discriminant analysis functions, which specify the key resilience predictors and their weights. The discriminant functions derived from this method can be applied to different study areas or times as long as the statistical assumptions are met [5]. More importantly, the RIM model uses empirical data to validate the resilience index, which is a critical property for a scientific index [25]. The RIM model has been applied to evaluate coastal-resilience

capacity of the counties along the Gulf of Mexico [5], the census block-group communities in the Lower Mississippi River Basin [10], and the Caribbean countries [21]. The RIM model has also been applied to evaluate the drought resilience of counties in the South Central USA [26,27] and earthquake resilience in China [28].

### 2.2. Social Media in Disaster Research

Social-media data present a new avenue for disaster-resilience research [18,19,29]. Numerous users can exchange information with each other anytime at any place through social-media platforms, making every user an active sensor of the environment. Some important resilience variables that are not accessible in traditional data sources could be obtained from social media, such as hazard acknowledgement, public sentiment, and social participation. As a result, many scholars have investigated social-media use in disasters and attempted to leverage social media to improve disaster resilience [20,30,31].

Dufty [32] developed a framework for analyzing the social aspects of disaster resilience-building. He suggested that social media can help reduce disaster risks, improve early warning communications, and provide social support in post-disaster recovery. Keim and Noji [33] studied the emergent use of social media in disaster management and found that opportunities and mechanisms for public participation in disaster relief are expanding because social media allows for widescale interaction between members of the public. This interaction can become collectively resourceful and generate information not otherwise readily obtainable through traditional disaster-management systems. Houston et al. [34] identified through a literature review that social media has at least 15 important resilience-enhancing functions from pre-event to post-event (e.g., signaling and detecting disasters, sending and receiving requests for help or assistance, documenting and learning about the disaster events, etc.). Redshaw et al. [35] found that the social cohesion index (CSI) was positively associated with both the extent of network communication and the level of community participation, suggesting that enhancing both network communication and community participation will result in higher levels of perception of social cohesion.

Several researchers have studied social-media responses specific to hurricanes. For example, Freberg et al. [31] analyzed social-media crisis messages through various qualitative and quantitative modeling techniques and generated a simple baseline model on what constitutes a "good" crisis message based on six aspects (response time, credibility, accuracy, simplicity, completeness, and broad communication). Using Hurricane Irene (2011) as a case study, they ranked effective and ineffective messages and identified a set of best practices for crisis communicators and reputation management (e.g., proper use of hashtags, and balance between official and conversational updates). Zou et al. [18,19] built a framework for Twitter collection and processing, defined Twitter indexes, and visualized and analyzed the disparities of Twitter use during Hurricanes Sandy and Harvey. The results support that social-media data can help in disaster recovery, emergency management, and resilience prediction. Wang et al. [36] studied the social disparities in Twitter use during Hurricane Sandy and found that socially vulnerable communities were digitally left behind in pre-disaster responses to Hurricane Sandy. Mihunov et al. [37] examined Twitter use in emergency rescue during Hurricane Harvey. They conducted an online survey of 195 respondents in Houston and found that Twitter use could be helpful in connecting flood victims with the first responders, indicating the potential of Twitter use for disaster rescue and management. Sadri et al. [38] analyzed Twitter data to understand information spreading during Hurricane Sandy. They found that information sharing follows a power-law distribution, suggesting the existence of few highly active nodes (users) in disseminating information, with many other nodes being less active. Their findings provide insights into how to account for user characteristics and network properties to spread information or limit the spreading of misinformation during hazards.

However, utilizing social-media data to assess, monitor, and enhance disaster resilience through quantitative research remains a challenge [17,39]. First, the volume of social-media data is huge, making the collecting and processing of such datasets time-

consuming and computationally challenging. Second, social-media data have many inherent issues, such as false information, malicious use, and lack of validation. Third, social-media users are not a representative sample of the population, with a tendency to skew toward young and urban individuals [40]. The biased demographic composition of users will contribute to bias in the study results. Fourth, algorithms designed to remove noise from social-media data may stand the danger of eliminating some valuable information [18]. Finally, information on the spatial location of users may be too coarse or unavailable, making it difficult to conduct accurate spatial analysis. Thus, it is necessary to learn how social media can be mined efficiently and correctly and then integrated with traditional data to gain a better understanding of disaster resilience.

### 2.3. Hurricane Isaac

Hurricane Isaac was a deadly and destructive tropical cyclone that came ashore in the U.S. state of Louisiana during August 2012 [41]. Hurricane Isaac made two landfalls, both at the same intensity (wind speed: 130km/h). It landed first on the coast of Louisiana near Plaquemines, LA, during the late evening hours of August 28, and then on the west of Port Fourchon, LA, in the early morning hours of August 29 (Figure 1). The inner core of Isaac moved slowly across central Louisiana on August 29 and 30, causing prolonged wind, coastal flooding, and flash-flooding. Early on 1 September 2012, Isaac dissipated to the west-southwest of Jefferson City, MO [41].

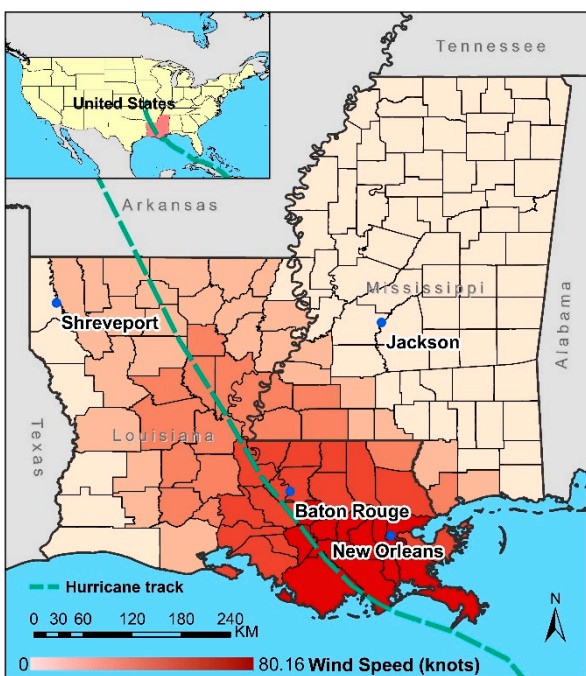

**Figure 1.** Hurricane Isaac track and study area.

According to the tropical cyclone's records from NOAA [41], Hurricane Isaac caused extensive storm surge and inland flooding over southern Mississippi and southeastern Louisiana, and it was directly responsible for 34 deaths and USD 2.35 billion in economic losses. Due to the huge damages, Louisiana and Mississippi received USD 329,715,102.55 and USD 37,399,646.71 of assistance, respectively, from FEMA for post-disaster recovery and reconstruction.

This study analyzes the disaster-related Twitter-use patterns during Isaac and how they correlate with the community resilience of counties impacted by Hurricane Isaac in Louisiana and Mississippi. The study area includes all 146 counties (99 of them have damage data) in the two states so that detailed regional analysis of Twitter activities and their implications to resilience during Hurricane Isaac can be conducted.

## 3. Data and Methods

### 3.1. Resilience Inference Measurement (RIM) Model

We applied the RIM model to evaluate the resilience of the 146 counties impacted by Hurricane Isaac. The RIM model defines resilience by three dimensions (hazard threat, damage, and recovery), which form two relationships—vulnerability (between hazard threat and damage) and adaptability (between damage and recovery) (Figure 2). Based on the level of hazard threat, damages endured, and rates of recovery after disastrous events, the RIM model categorizes communities into four resilience levels; from low to high they are susceptible, recovering, resistant, and usurper. Their corresponding resilience scores are from 1 to 4 [5]. In general, susceptible communities are characterized by high vulnerability (high damage relative to the hazard threat) and low adaptability (low recovery, such as return of population), whereas usurper communities are characterized by low vulnerability and high adaptability.

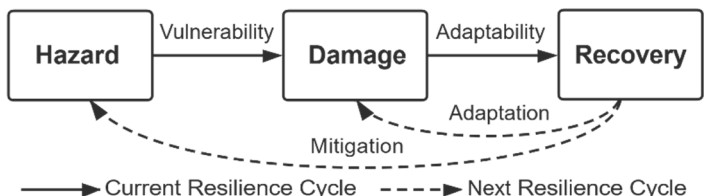

**Figure 2.** The resilience inference measurement (RIM) framework (modified from Lam et al. [42]).

The calculation of the RIM scores consists of two steps: K-means clustering for initial grouping, and discriminant analysis for validating the grouping and extracting important socioeconomic indicators ([10] Figure 3). K-means analysis is a nonhierarchical clustering method. It aims to partition observations or cases into "k" groups, where each case is assigned to the cluster that has the nearest distance to its centroid. This clustering process allows a classification of four resilience types based on the similarity of the three dimensions.

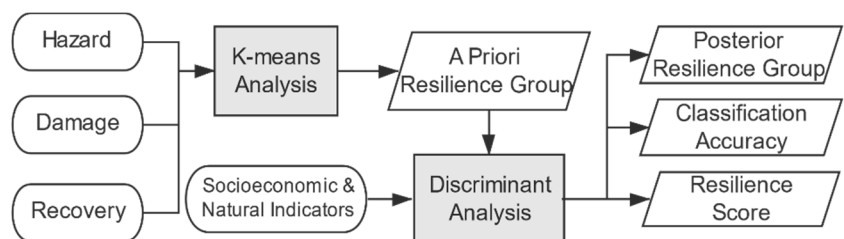

**Figure 3.** Framework of the RIM model (modified from Cai et al. [10]).

Once the a priori groups are determined, discriminant analysis is used to test whether these a priori groups can be characterized by a set of pre-disaster resilience-capacity indicators. Discriminant analysis is an inferential statistical technique that is used when the dependent variable is categorical, and the independent variables are interval or ratio. In the RIM model, the categorical variable is the level of resilience (susceptible, recovering, resistant, usurper; from 1 through 4, respectively) derived from k-means, the independent variables are the social–environmental variables related to resilience-capacity building. Discriminant analysis outputs include the classification accuracy between the a priori group classification and the posterior group classification, discriminant functions containing the coefficients (weights) of major resilient variables for predicting the group membership of an observation (county), and the probabilities of each county belonging to each of the four groups.

Based on the probabilities of group membership, the discrete resilience level of each observation can be converted to a continuous RIM score using the following equation:

$$RIMscore = \sum_{i=1}^{m} i \times Prob(i) \tag{1}$$

where *m* is the number of resilience groups (i.e., 4), and *Prob(i)* is the posterior probability of an observation belonging to resilience group *i* [5,10,43]. In sum, the RIM model is an inferential technique because the discriminant functions derived can be used to estimate the resilience level of communities in other study areas or over time when the statistical assumptions are met. Furthermore, the RIM functions and scores are already validated because of the use of real data on hazard threat, damage sustained, and recovery rate in the derivation process.

In this study, hazard threat is represented by the average wind speed in each county, the data of which were obtained from the World Meteorological Organization [44]. There were 51 records along the hurricane track for Isaac, each with a time stamp, latitude, longitude, wind speed in knots (kt), and atmospheric pressure in millibars (mb). These point data were converted into a hazard-threat-level surface using the Kernel Density tool in ArcGIS. In Kernel Density, the output cell size was set to 96.5 meters, and the search radius was 200 km, which is the radius of Hurricane Isaac. After the Kernel Density operation, a raster with 9312.25 m$^2$ pixel size was generated and the Zonal Statistic tool was used to derive the average wind speed by county.

Damage data were obtained from the Storm Events Database from the National Climatic Data Center, NOAA [45]. The original file had 64,503 records of storm events from 2012 to 2017. Among them, 85 and 88 damage records in Louisiana and Mississippi, respectively, were associated with Hurricane Isaac. In total, 99 counties had damage data, and 47 counties did not.

Recovery rate is represented by population change from 2011 to 2013. Population data by county were obtained from the U.S. Census Bureau (see https://data.census.gov/cedsci/advanced). Considering a hurricane as an event, its impact on population change could be short-term. It is reasonable to observe the population change between 2011 and 2013 as a proxy reflecting the recovery process in a short time period.

A total of 30 demographic, social-capital, and economic indicators were selected as potential resilience-capacity indicators. These variables were selected according to previous studies [5,10,18,26]. Of the 30 variables, 26 of them were census variables of 2010 (https://data.census.gov/cedsci/advanced). The variable on percent area in the FEMA flood zone was calculated by the authors using the raw data from the FEMA National Flood Hazard Layer in 2014 (FEMA 2014) (https://msc.fema.gov/portal/advanceSearch). Variables on percent area covered by water and percent impervious area in 2001 were originally from the National Land Cover Database (Multi-Resolution Land Characteristics Consortium, 2001) (www.mrlc.gov/data) in the form of 30 m pixels, and elevation data in 2000–2007 were downloaded from the U.S. Geological Survey National Elevation Dataset (USGS 2001) (http://nationalmap.gov/elevation.html), also in the form of 30 m pixels. These three variables were tabulated into the county level and provided for this study by Dr. Heng Cai [10].

### 3.2. Twitter Data Collection and Preprocessing

Hurricane Isaac originated on 16 August 2012, strengthened to a real threat to the southern coast of the United States on August 21, made the first landfall on August 28, and disintegrated on September 1. This research collected Twitter data using the two keywords "hurricane" and "Isaac" from August 21 when Isaac was formed to September 17, two weeks after the hurricane dissipated. This complete Twitter dataset was purchased from Gnip Inc. (Boulder, CO, USA), a social-media data-aggregation company affiliated with Twitter. The dataset had a total of 2,867,776 records (tweets) in the JavaScript Object Notation (JSON) format. Each record contains the time of the tweet, tweet content, user profile information, and possibly locational information such as coordinates and places of

the tweet. The time of the tweet was used to divide the data into preparedness, response, and recovery phases. We defined the week before Isaac's landfall (21–26 August 2012) as the preparedness phase, August 27–31 as the response phase, and September 1–17 as the recovery phase. A python script was created and used to parse, classify, geocode, and store Twitter records into a Mongo database.

To determine every tweet's location, information from the user's profile was extracted if there was no real-time location in the tweet; otherwise, the x-y coordinate pairs were collected and saved directly. User profile location is usually recorded as a point of interest, a street, a neighborhood, or a city, which can be converted to a point with coordinates through geocoding. Graham et al. (2014) found that in their 4000 randomly selected tweets, 54.5% of them were associated with a city name and could be correctly located by Google's geocoding service. We implemented Nominatim, an open-source geocoding service that can be deployed on a local server and is powered by OpenStreetMap georeferencing data (https://www.openstreetmap.org/). We were able to assign locations to 1,686,851 Tweets out of the total of 2,867,776 (59.82%).

### 3.3. Twitter Indices

Tweet density by county for all three phases and the entire period were calculated by dividing the number of tweets in each county by its 2012 population (Equation (1)). Previous studies have shown that tweet density during disasters can reflect the geographic and demographic characteristics of communities [18,19]. Communities with higher frequencies of disaster-related tweets are usually characterized by a high proportion of young people and proximity to the epicenter of disaster [46]. Tweet density was also used to indicate situational awareness [20], risk perception [47], or level of concern [48] in other studies.

$$Tweet\ Density = \frac{\#Tweets}{Population 2012} \qquad (2)$$

Since higher Twitter use in a community could be due to its proximity to the disaster location, this study will analyze the correlation between tweet density and hazard threat level to determine if proximity to the location of disaster plays a role in Twitter use density.

Sentiment analysis of a tweet's content (i.e., characterizing the words in a tweet as positive or negative) has been used to indicate real-time attitude from the public toward an event [18,19,49]. However, it is not known if sentiment expressed in the tweets correlated with community resilience. As mentioned above, we hypothesize that communities of higher resilience capacity generally have more positive Twitter responses during disasters. In this study, we used VADER (Valence Aware Dictionary for Sentiment Reasoning), a commonly used lexicon and rule-based python module, to assign a sentiment score for each tweet. The score ranges from –1 to +1, with –1 being the most negative, and +1 the most positive. By using geocoding and spatial analysis, the mean sentiment scores were tabulated for each county for the entire period and all three phases.

In sum, the analyses include three steps. First, we used the RIM model to calculate the resilience score for each of the 146 counties in the study area. Second, Twitter indices including tweet density and sentiment score were calculated for each county for all three phases. Third, Pearson correlation analysis was conducted to test if the two sets of variables were correlated.

## 4. Results

### 4.1. Disaster-Resilience Scores

Figure 4 shows the spatial patterns of the three dimensions (hazard threat, damage, and recovery) and the final RIM score. Table 1 lists the summary statistics of key variables, including tweet density (TD) and sentiment score (SS). K-means clustering based on the three dimensions generated four resilience groups. This initial group designation, together with the 30 social–environmental variables (Table 2), were input to discriminant analysis. We used the stepwise option to select significant predictor variables while minimizing the

collinearity among the selected variables. Six variables were selected, including percent households with telephone available, median income, number of households per km², percent households with no vehicle available, percent impervious area, and percent area covered by water. Based on the derived discriminant model, discriminant analysis recalculated the level of resilience for each county. The classification accuracy between the initial groups from K-means and the posterior groups from discriminant analysis was 75.8%, and the accuracy of the leave-one-out cross-validated results was 71.7%, which supports the robustness of the model (Table 3). Finally, the continuous RIM scores were calculated based on the probabilities of each county belonging to each of the four groups (Equation (1)).

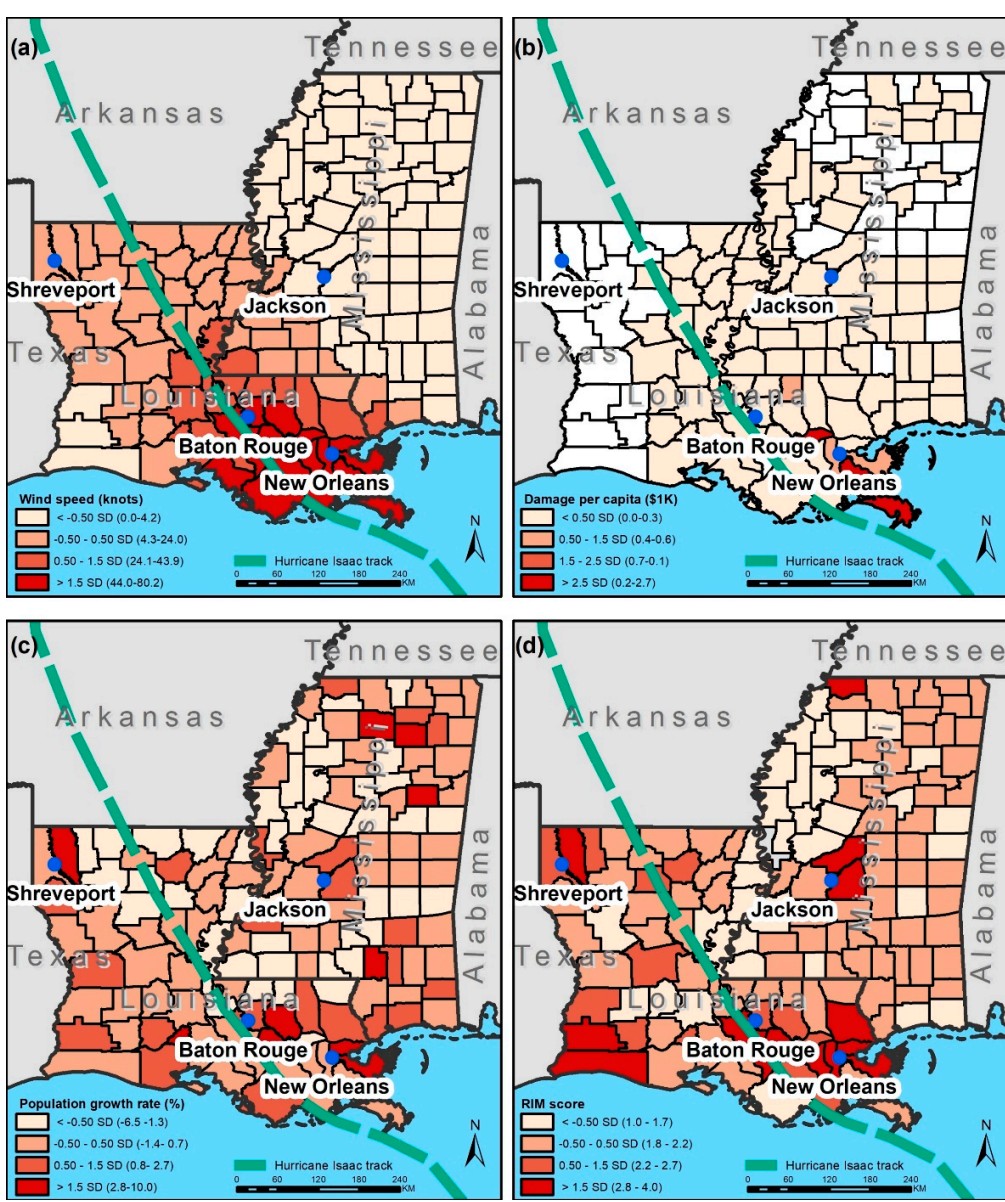

**Figure 4.** County-level map of (**a**) hazard threat level (average wind speed in knots); (**b**) Damage per capita; (**c**) Recovery (as represented by population growth rate for 2010–2013); and (**d**) the RIM scores. SD = standard deviation.

**Table 1.** Descriptive statistics of major indicators.

|  | N | Min | Max | Mean | Std. Dev. |
|---|---|---|---|---|---|
| **Windspeed (knots)** | 144 | 0.00 | 80.16 | 14.13 | 19.96 |
| **Damage per capita (1k USD)** | 98 | 0.00 | 2.67 | 0.09 | 0.36 |
| **Population growth rate** | 144 | −6.50 | 9.95 | −0.26 | 2.01 |
| **% households, telephone av.** | 144 | 75.42 | 98.36 | 93.88 | 3.17 |
| **# households per km$^2$** | 144 | 0.44 | 303.25 | 14.45 | 30.96 |
| **% households with no vehicle** | 144 | 2.48 | 18.22 | 8.18 | 3.29 |
| **Median income** | 144 | 21,360.00 | 63,716.00 | 36,659.00 | 8841.00 |
| **% area covered by water** | 144 | 0.00 | 0.74 | 0.06 | 0.13 |
| **% impervious area** | 144 | 0.10 | 21.95 | 1.62 | 2.59 |
| **RIM score** | 144 | 1.09 | 4.00 | 1.96 | 0.49 |
| **TD$_{Preparedness}$** | 144 | 0.00 | 0.02 | 0.00 | 0.00 |
| **TD$_{Response}$** | 144 | 0.00 | 0.12 | 0.01 | 0.01 |
| **TD$_{Recovery}$** | 144 | 0.00 | 0.03 | 0.00 | 0.00 |
| **TD$_{All}$** | 144 | 0.00 | 0.17 | 0.01 | 0.02 |
| **SS$_{Preparedness}$** | 139 | −0.65 | 0.56 | 0.04 | 0.16 |
| **SS$_{Response}$** | 143 | −0.14 | 0.39 | 0.06 | 0.07 |
| **SS$_{Recovery}$** | 139 | −0.63 | 0.60 | 0.03 | 0.20 |
| **SS$_{All}$** | 143 | −0.17 | 0.31 | 0.04 | 0.09 |

TD = Tweet density; SS = sentiment score.

**Table 2.** The 30 predictor variables (2010) used in discriminant analysis.

| Category | Variables | Source |
|---|---|---|
| Social | % population over 65 years old | U.S Census Bureau |
|  | % population less than 9th grade | U.S Census Bureau |
|  | % female householder | U.S Census Bureau |
|  | # households per km$^2$ | U.S Census Bureau |
|  | % household no vehicle available | U.S Census Bureau |
|  | Median age | U.S Census Bureau |
|  | Average household size | U.S Census Bureau |
|  | % people voted | U.S Census Bureau |
| Economic | % natural res., constr., maint. occupations | U.S Census Bureau |
|  | % agriculture employment | U.S Census Bureau |
|  | Household median income | U.S Census Bureau |
|  | Median value of housing | U.S Census Bureau |
|  | Median rent | U.S Census Bureau |
|  | % civilian labor force | U.S Census Bureau |

**Table 2.** *Cont.*

| Category | Variables | Source |
|---|---|---|
| Community | % people born in US | U.S Census Bureau |
| | % household below poverty level | U.S Census Bureau |
| | Employed rate | U.S Census Bureau |
| | Unemployment rate | U.S Census Bureau |
| | % pop. over 25 with bachelor's degree | U.S Census Bureau |
| | % people with a disability | U.S Census Bureau |
| | % federal exp. on disability and retirement | U.S Census Bureau |
| Infrastructure | % mobile homes and all other types of units | U.S Census Bureau |
| | % housing built 2000 and later | U.S Census Bureau |
| | # healthcare providers per capita | U.S Census Bureau |
| | % urban housing units in whole county | U.S Census Bureau |
| | % households with telephone available | U.S Census Bureau |
| Environment | Mean elevation (meters) | National Elevation Dataset |
| | % area in FEMA flood zone | National Flood Hazard Layer |
| | % area covered by water | National Land Cover Database |
| | % impervious area | National Land Cover Database |

**Table 3.** Classification results from k-means and discriminant analyses.

| K-Means Analysis | Discriminant Analysis | | | | |
|---|---|---|---|---|---|
| | Susceptible | Recovering | Resistant | Usurper | Total |
| Susceptible | 4 (57.1%) | 2 | 1 | 0 | 7 |
| Recovering | 10 | 59 (79.7%) | 5 | 0 | 74 |
| Resistant | 3 | 3 | 10 (62.5%) | 0 | 16 |
| Usurper | 0 | 0 | 0 | 2 (100%) | 2 |
| Ungrouped | 10 | 31 | 6 | 0 | 47 |
| Total | 27 | 95 | 22 | 2 | 146 |

The overall classification accuracy was 75.8%. The leave-one-out cross-validated classification accuracy was 71.7%.

The map of hazard threats shows a gradual decline from the coast to inland, as expected. On the contrary, the damage map exhibits a rather drastic pattern, with a few counties along the coast (near New Orleans) having very high damages (above 2.5 standard deviations from the mean), while most of the remaining counties had below-average damages. There was no discernable trend regarding the recovery (population growth) pattern, although generally the counties experiencing higher population growth were either proximal to major universities (such as Oxford, Starkville, and Harrisburg in Mississippi) or major cities. The RIM analysis shows that the two highest-resilience (usurper) counties were in the New Orleans area. Counties surrounding major cities such as Baton Rouge, Jackson, and Shreveport also had higher resilience. Lower-resilience counties were mostly found in inland rural areas, except for Harrison, MS (which contains the cities of Gulfport and Biloxi), located along the Gulf Coast.

*4.2. Spatial–Temporal Patterns of Tweet Density and Sentiment*

Figure 5 shows the total number of tweets daily and Table 4 shows the number of geocoded tweets in each of the three phases. As expected, the discussion of Hurricane

Isaac started to grow slowly from August 21 to 25, then it increased sharply from August 25 to August 27, three days before the landfall, indicating the growing public awareness. The highest number of tweets was observed on 29 August 2012, the day of the second landfall. Discussions dissipated on August 30 and 31 along with the fading hurricane. In the following 17 days of September, there was no significant discussion of Hurricane Isaac.

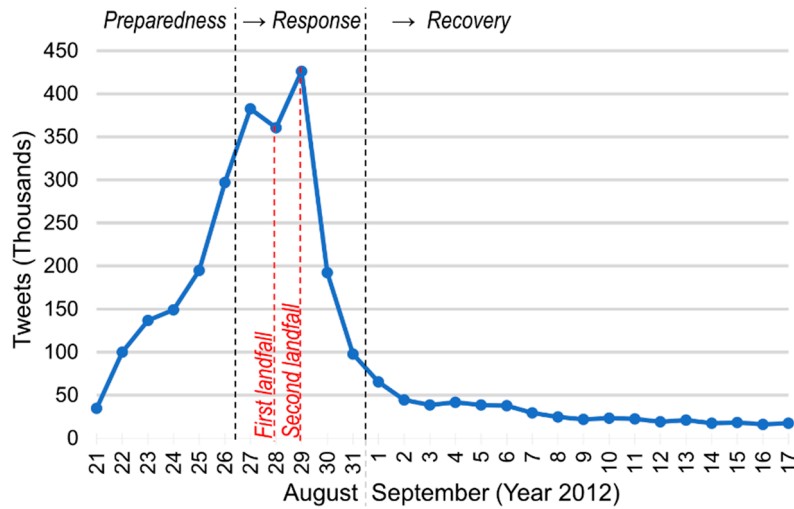

**Figure 5.** Daily Hurricane Isaac tweet totals (black dashed lines separate emergency-response phases).

**Table 4.** Number of geocoded tweets and all tweets in each phase during Hurricane Isaac.

|  | Preparedness | Response | Recovery | Total |
|---|---|---|---|---|
| **Geocoded Tweets** | 552,742 | 852,433 | 281,605 | 1,686,851 |
| **All Tweets** | 912,035 | 1,459,252 | 496,491 | 2,876,726 |

Figure 6 maps Twitter use patterns of the three phases and all time. We excluded two counties, Avoyelles, LA, and Attala, MS, in the descriptive statistics and correlation analysis because of the imprecise locational specification in the user profiles. Both Avoyelles, LA, and Attala, MS, were found to have very high values of tweet density, even though they are in rural areas. By examining the tweets' location information and location of the two counties in their states, we found that many people did not provide detailed location information in either their tweets or profiles, they only put "Louisiana" or "Mississippi" for the location information. The tweets containing the state name only as the location were geocoded to the centers of the states, which fall into the two counties. Therefore, to eliminate the effect of the improper values, we excluded the two counties in the analysis.

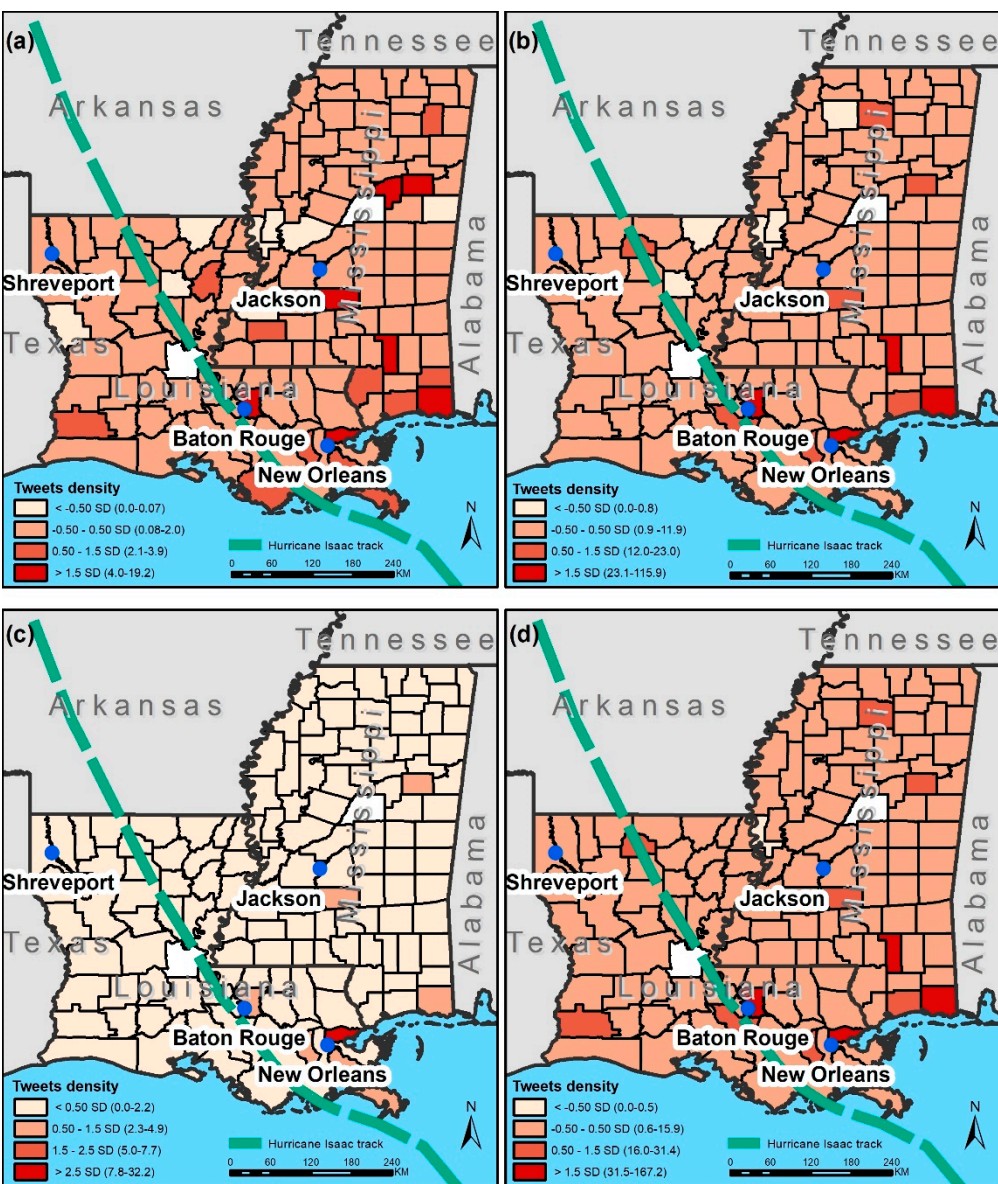

**Figure 6.** Spatial patterns of tweet density (per thousand people) in (**a**) Preparedness (**b**) Response (**c**) Recovery phases and (**d**) All time. SD = standard deviation.

In general, more counties tweeted with higher density during the preparedness phase, which may be due to the uncertainty of the location of the landfall. During the response phase (after the landfall), counties to the east of the hurricane track, especially those located along the coast, tweeted more, which could be explained by the fact that places east of a hurricane track often sustain far more damage than places west of the track. Twitter use declined during the recovery phase, with only a few counties having a high tweet density. Overall, for the entire period, coastal counties east of the track tweeted more, and for inland counties that tweeted more, they were located near major universities or cities.

Figure 7 maps the mean sentiment scores by county in three phases and for all time (some of the counties do not have a sentiment score because they had no tweets). Unfortunately, the spatial patterns of sentiment scores appeared random; they did not show any discernable trend spatially or temporally. The lack of correlation between sentiment scores and the resilient variables, as shown below, supports this observation.

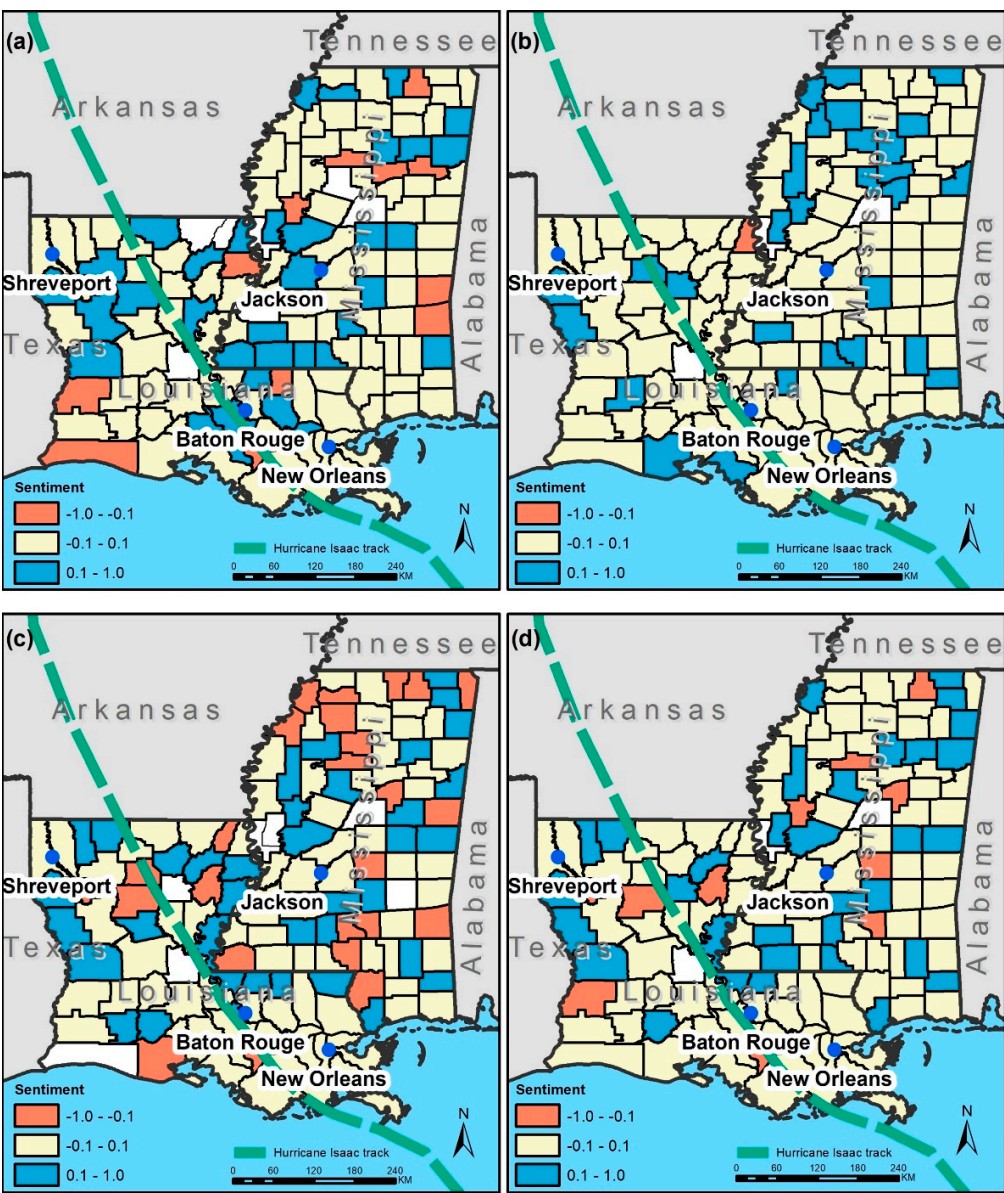

**Figure 7.** Spatial patterns of sentiment scores in (**a**) Preparedness (**b**) Response (**c**) Recovery phases and (**d**) All time.

To provide a context, Table 5 lists the top five negative and positive tweets related to Hurricane Isaac. According to the content of the tweets, the most negative tweets were the ones mentioning the damage, such as housing damage, injury, and inconvenience regarding leisure time. Tweets considered most positive were related to well-wishing and praying for others, and the users associated with these tweets were not affected by the disaster.

**Table 5.** The top five Isaac-related tweets with lowest and highest sentiment.

| Rank | Tweet Content | Scores |
|:---:|:---:|:---:|
| | *Top 5 negative tweets* | |
| 1 | &lt;expletive&gt; Tropical/Hurricane Issac &lt;expletive&gt; Bobby Jindal pdn &lt;expletive&gt; Romney &lt;expletive&gt; the police & &lt;expletive&gt; u hating &lt;expletive&gt; &lt;expletive&gt; & &lt;expletive&gt; &lt;expletive&gt; &lt;expletive&gt;!!! | −0.9881 |
| 2 | &lt;expletive&gt; Romney, &lt;expletive&gt; that 12*21*12 BS, &lt;expletive&gt; Sprint, &lt;expletive&gt; Hurricane Isaac tht did 2 Billion in damage, &lt;expletive&gt; Summer and &lt;expletive&gt; dead beat dads. | −0.9826 |
| 3 | RT @&lt;user&gt;: &lt;EXPLETIVE&gt; THIS &lt;expletive&gt; &lt;expletive&gt; HURRICANE IF YOU COME THRU & RUIN MY WEEKEND I'LL KILL YOUR FAMILY | −0.9795 |
| 4 | &lt;expletive&gt; all this stupid hurricane &lt;expletive&gt; It's not like Oma get hurt any worse &lt;expletive&gt; this &lt;expletive&gt; #drove | −0.9781 |
| 5 | SO WORRIED ABOUT MY ❤'S N NEW ORLEANS I HATE THIS HURRICANE :( ITS GONNA RUIN MY VERY PLANNED MUCH NEEDED VACATION I HATE MI :( | −0.9755 |
| | *Top 5 positive tweets* | |
| 1 | HA HA HA HA HA HA RIGHT RT @BreakingNews VP Biden to skip stop in Tampa to ensure resources are not distracted as Isaac approaches | 0.9788 |
| 2 | I HOPE GOD PROTECTS MY BEST FRIEND THAT LIVEs ON THE COAST WHEN THAT HURRICANE COMES I ALSO HOPES HE PROTECTS ME MY FAMILY & FRIENDS ALSO ! | 0.9782 |
| 3 | FLORIDA URGENT-HELP NEEDED TO KEEP RESCUE DOGS SAFE FROM TROPICAL STORM ISAAC! SHARE SHARE!! If &lt;link&gt; | 0.9772 |
| 4 | Good Morning ERRRBODY! THANK GOD FOR ANOTHER BLESSED DAY! PLEASE BE SAFE N PRAY FOR US HERE N FLORIDA HAITI DOMIN REP ! #Isaac! | 0.9763 |
| 5 | Prayers And Blessings For The Good People Of New Orleans And Gulf Coast For Oncoming Hurricane BE SAFE BE WELL Peace And Love God Bless You" | 0.9763 |

In addition, we plotted the average sentiment scores in three phases and compared those of Hurricanes Sandy and Harvey (Figure 8; [19]). Isaac-sentiment and Harvey-sentiment shared the same pattern, with more neutral sentiment expressed at the beginning. Sentiment became more positive during the response phase, but it dropped sharply toward more negative during the recovery phase. There could be two reasons for Isaac's sentiment trend. First, Louisiana and Mississippi declared states of emergency on the afternoon of August 26, which was less than one day before the landfall, so residents were underprepared for this disaster. During the response phase, Twitter users shared disaster-related information, generally omitting the emotionally charged words, which could have led to higher sentiment scores. Finally, up until it dissipated on September 2, Hurricane Isaac brought tornados, floods, and damage to the roads and buildings, which made for a bitter aftermath and difficult recovery, thus the tweets tended to be negative and emotional, leading to a low sentiment score.

*4.3. Correlations between Twitter Indices and Disaster Resilience*

Pearson correlation analyses were conducted between Twitter indices (tweet density and sentiment score) and all 30 predictor variables to test the hypothesis that higher-resilience counties were associated with higher Twitter use and more positive sentiment scores. Table 6 lists the correlation results of major resilience variables only (the three elements, the six variables derived from the stepwise discriminant analysis, and the RIM score). A full correlation table is listed in Appendix A.

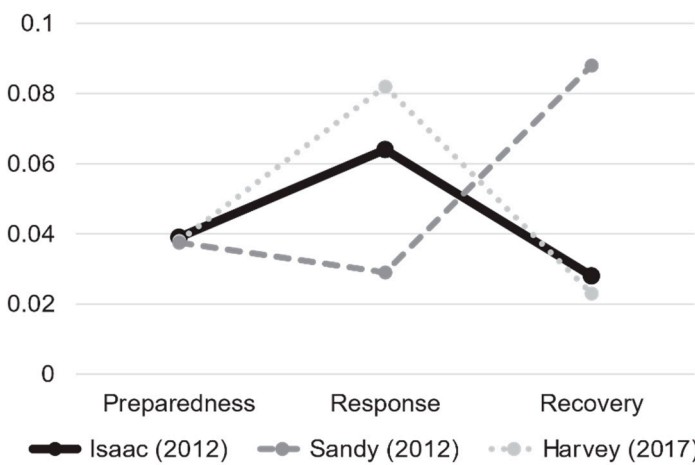

**Figure 8.** Mean sentiment scores during Hurricanes Isaac, Sandy, and Harvey.

**Table 6.** Tweet density correlations (selected listing).

| | RIM Score | $TD_{Preparedness}$ | $TD_{Response}$ | $TD_{Recovery}$ | $TD_{All}$ | $SS_{Preparedness}$ | $SS_{Response}$ | $SS_{Recovery}$ | $SS_{All}$ |
|---|---|---|---|---|---|---|---|---|---|
| Windspeed (knots) | 0.505 ** | 0.223 ** | 0.258 ** | 0.259 ** | 0.259 ** | 0.024 | −0.103 | −0.004 | −0.018 |
| Population growth rate | 0.533 ** | 0.320 ** | 0.303 ** | 0.265 ** | 0.304 ** | −0.012 | −0.056 | 0.038 | −0.008 |
| % hh, teleph. av. | 0.254 ** | −0.077 | −0.056 | −0.068 | −0.062 | 0.063 | −0.162 | 0.148 | 0.100 |
| Households per $km^2$ | 0.488 ** | 0.728 ** | 0.787 ** | 0.816 ** | 0.799 ** | −0.004 | −0.004 | 0.016 | 0.007 |
| % household, no vehicle | −0.439 ** | 0.022 | 0.060 | 0.116 | 0.066 | −0.073 | −0.025 | −0.070 | −0.100 |
| Median income | 0.701 ** | 0.139 | 0.133 | 0.089 | 0.129 | 0.048 | −0.053 | 0.052 | 0.040 |
| % area covered by water | 0.411 ** | 0.325 ** | 0.312 ** | 0.332 ** | 0.323 ** | −0.045 | −0.119 | −0.045 | −0.100 |
| % impervious area | 0.554 ** | 0.631 ** | 0.680 ** | 0.704 ** | 0.691 ** | −0.014 | −0.020 | 0.007 | −0.011 |
| RIM score | 1.000 | 0.312 ** | 0.331 ** | 0.374 ** | 0.342 ** | 0.043 | −0.077 | 0.035 | 0.022 |
| $TD_{Preparedness}$ | 0.312 ** | 1.000 | 0.936 ** | 0.903 ** | 0.955 ** | −0.004 | −0.026 | −0.032 | −0.037 |
| $TD_{Response}$ | 0.331 ** | 0.936 ** | 1.000 | 0.918 ** | 0.996 ** | −0.005 | −0.062 | −0.007 | −0.030 |
| $TD_{Recovery}$ | 0.374 ** | 0.903 ** | 0.918 ** | 1.000 | 0.947 ** | −0.014 | −0.025 | 0.012 | −0.008 |
| $TD_{All}$ | 0.342 ** | 0.955 ** | 0.996 ** | 0.947 ** | 1.000 | −0.006 | −0.053 | −0.007 | −0.027 |
| $SS_{Preparedness}$ | 0.043 | −0.004 | −0.005 | −0.014 | −0.006 | 1.000 | −0.115 | −0.016 | 0.571 ** |
| $SS_{Response}$ | −0.077 | −0.026 | −0.062 | −0.025 | −0.053 | −0.115 | 1.000 | 0.027 | 0.256 ** |
| $SS_{Recovery}$ | 0.035 | −0.032 | −0.007 | 0.012 | −0.007 | −0.016 | 0.027 | 1.000 | 0.775 ** |
| $SS_{All}$ | 0.022 | −0.037 | −0.030 | −0.008 | −0.027 | 0.571 ** | 0.256 ** | 0.775 ** | 1.000 |

** Correlation was significant at the 0.01 level (2-tailed). * Correlation was significant at the 0.05 level (2-tailed).

Tweet density of each of the three phases and the entire period were found to correlate positively ($p < 0.01$) with three of the resilience predictors and the RIM score. The three resilience predictors were number of households per $km^2$, percent area covered by water, and percent impervious area. These three variables were considered as the "environmental" characteristics of the counties and could be interpreted as a factor of the degree of urban development. Surprisingly, the other three resilience variables that were related to the socioeconomic characteristics of the counties—median income, percent of households with telephone available, and percent of households with no vehicle available—did not have significant correlation with tweet density. However, the combined resilience capacity of counties as represented by the RIM scores had significant positive correlation with the tweet density of all phases and the entire period. A closer look at the full correlation table shows that tweet density also had significant positive correlations with several socioeconomic variables that are not listed in Table 6, including, for example, percent population over 25 with a bachelor's degree, median housing value, and median rent. At the same time, tweet

density also had significant negative correlations with percent population over 65 years old, median age, and percent population who were native born. We can interpret that these variables had high collinearity among themselves and with the six variables, therefore they were not picked for the final RIM resilience calculation, but their variances had already been captured in the final RIM score. In summary, the correlation results showed that the higher the resilience score, the higher the use of Twitter during a disaster event. This finding supports the first part of our hypothesis. It also implies that geographical and social disparities in Twitter use existed during the disaster, which in turn could have affected the recovery effort of communities.

No significant correlations were found between sentiment score and resilience variables. This finding is new and surprising. The finding contradicts with the "conventional" belief and did not support the second part of our hypothesis, which specified that the higher the resilience score of a community, the more positive their tweet messages during a disaster event. In other words, we cannot use the sentiment score expressed in the tweets as a predictor of resilience in a community.

## 5. Discussion

The significance of this study is threefold. First, the study was driven by the research question, which is, will communities with more disaster-related Twitter uses be more resilient to disasters, presumably because they have better situational awareness? The underlying inquiry posits that if there are social and geographical disparities in Twitter use, how will such disparities affect communities' resilience to disasters? Several studies on the issue of geographical and social disparities of Twitter use for Hurricane Sandy have been conducted (e.g., [18,36]), however, there is no similar study on Hurricane Isaac, which occurred in the same year (2012). This study is among the first to explore the early use of Twitter during a disaster in a southern U. S. region. While direct comparisons between studies were difficult, the fact that the number of Isaac-related tweets (about 2.9 million) in a three-week period was far less than the number of tweets during Sandy (over 16 million tweets, also purchased by the authors) showed a distinct disparity not only within a region, but also across geographical regions. This could further imply that there were distinct differences in the level of disaster resilience across regions, and more research will be needed to improve regional resilience. Thus, our study of the early use of Twitter during a disaster provides rare, historical baseline information on the relationship between social media and disaster resilience.

Second, findings from this study add new knowledge and insights into the use of Twitter data for disaster-resilience analysis. The study results supported the first part of our hypothesis, that there was positive correlation between the RIM score and tweet density, thus implying that geographical and social disparities in Twitter use existed during Hurricane Isaac. However, the second finding, that no significant correlations existed between sentiment score and resilience variables, was new and surprising. The result contradicted the "traditional" belief, suggesting the need for more research on a better approach to detect sentiment and text mining to extract other useful information from the wealth of Twitter data.

Third, the study highlights two technical challenges that need to be tackled in future research. First, the location information extracted from the tweets may not be accurate because of the broad location specifications the users provided. For example, when a user specifies his/her location with only the state name (Louisiana or Mississippi), the tweet will fall into the centroid of the state. In this study, they happened to be in rural counties (Avoyelles, LA, and Attala, MS) and therefore were easily detected as unreasonable. In future studies, the centroids may fall into urban areas and will be difficult to detect and correct. More robust methods will need to be explored for more accurate location-information extraction. The second challenge refers to the sentiment analysis. In addition to the need for improving the sentiment characterization algorithm, more research is needed

to classify tweets based on their content to further explore the application potentials of tweets in predicting community resilience.

## 6. Conclusions

This study analyzed the correlations between Twitter use and community resilience using a case study of 146 counties in Louisiana and Mississippi affected by Hurricane Isaac in August 2012. We hypothesized that counties with higher resilience levels would be associated with higher tweet densities and more positive sentiment scores. We applied the resilience inference measurement (RIM) model to compute the resilience scores of all counties in the study area. A complete Twitter dataset containing 2,867,776 tweets extracted using two keywords of "hurricane" and "Isaac" from August 21 to 17 September 2012 was purchased. Twitter indices, including tweet density and sentiment score by county, were calculated for the three main phases of emergency management (preparedness, response, and recovery). Pearson's correlation analysis between the two sets of variables was conducted.

Significant positive correlations were found between tweet density and the RIM scores ($r > 0.31$, $p < 0.01$), as well as with other social–environmental indicators, confirming that communities with higher resilience capacity, which are characterized by better social-environmental conditions, tended to have higher Twitter use. These results imply that Twitter use during disasters could be improved to increase the resilience of affected communities. However, no significant correlations were found between sentiment scores and resilience measures, suggesting that further research on sentiment analysis, as well as other means of extracting useful information from the tweets, are needed.

**Author Contributions:** Conceptualization, Nina S. N. Lam and Kejin Wang; methodology, Nina S. N. Lam, Lei Zou; software, Kejin Wang and Lei Zou; validation, Nina S. N. Lam, Kejin Wang, Lei Zou, and Volodymyr Mihunov; formal analysis, Kejin Wang and Volodymyr Mihunov; investigation, Kejin Wang; resources, Nina S. N. Lam.; data curation, Kejin Wang; writing—original draft preparation, Kejin Wang; writing—review and editing, Nina S. N. Lam, Kejin Wang, Volodymyr Mihunov, and Lei Zou; visualization, Kejin Wang and Volodymyr Mihunov; supervision, Nina S. N. Lam; project administration, Nina S. N. Lam; funding acquisition, Nina S. N. Lam. All authors have read and agreed to the published version of the manuscript.

**Funding:** This research is based on work supported by two research grants from the U.S. National Science Foundation: The Interdisciplinary Behavioral and Social Science Research (IBSS) Program (Award No. 1620451) and Rapid Response Research (RAPID) Program (Award No. 1762600). Any opinions, findings, and conclusions or recommendations expressed in this article are those of the authors and do not necessarily reflect the views of the funding agency.

**Institutional Review Board Statement:** Not applicable.

**Informed Consent Statement:** Not applicable.

**Data Availability Statement:** Not applicable.

**Acknowledgments:** We acknowledge Heng Cai for providing the county-level elevation and land cover data. We would like to thank the anonymous reviewers and editors for providing valuable comments and suggestions.

**Conflicts of Interest:** The authors declare no conflict of interest.

## Appendix A

**Table A1.** Tweet-density correlations.

| | TD$_{Preparedness}$ | TD$_{Response}$ | TD$_{Recovery}$ | TD$_{All}$ | RIMscore | SS$_{Preparedness}$ | SS$_{Response}$ | SS$_{Recovery}$ | SS$_{All}$ |
|---|---|---|---|---|---|---|---|---|---|
| Wind speed (knots) | 0.223 ** | 0.258 ** | 0.259 ** | 0.259 ** | 0.505 ** | 0.024 | −0.103 | −0.004 | −0.018 |
| Damage per capita | 0.061 | 0.033 | 0.089 | 0.047 | 0.203 * | 0.000 | −0.093 | −0.074 | −0.077 |
| Pop. growth rate | 0.320 ** | 0.303 ** | 0.265 ** | 0.304 ** | 0.533 ** | −0.012 | −0.056 | 0.038 | −0.008 |
| % over 65 | −0.191 * | −0.230 ** | −0.161 | −0.217 ** | −0.283 ** | 0.013 | 0.066 | −0.034 | 0.006 |
| % hh, with teleph. | −0.077 | −0.056 | −0.068 | −0.062 | 0.254 ** | 0.063 | −0.162 | 0.148 | 0.100 |
| % hous. in urb. area | 0.300 ** | 0.352 ** | 0.264 ** | 0.337 ** | 0.420 ** | −0.068 | −0.120 | 0.019 | −0.065 |
| % pop. below 9th grade | −0.250 ** | −0.246 ** | −0.162 | −0.236 ** | −0.460 ** | −0.080 | 0.093 | −0.041 | −0.045 |
| % female householder | −0.073 | −0.012 | 0.013 | −0.015 | −0.393 ** | −0.058 | −0.026 | −0.065 | −0.076 |
| Households per km$^2$ | 0.728 ** | 0.787 ** | 0.816 ** | 0.799 ** | 0.488 ** | −0.004 | −0.004 | 0.016 | 0.007 |
| % hh, no vehicle | 0.022 | 0.060 | 0.116 | 0.066 | −0.439 ** | −0.073 | −0.025 | −0.070 | −0.100 |
| % hh below poverty | −0.111 | −0.074 | −0.054 | −0.077 | −0.656 ** | −0.060 | 0.023 | −0.047 | −0.052 |
| Median income | 0.139 | 0.133 | 0.089 | 0.129 | 0.701 ** | 0.048 | −0.053 | 0.052 | 0.040 |
| Median housing value | 0.372 ** | 0.383 ** | 0.332 ** | 0.380 ** | 0.632 ** | 0.015 | −0.051 | 0.060 | 0.031 |
| % min-ing/constr/maint | −0.090 | −0.140 | −0.118 | −0.132 | 0.132 | −0.059 | 0.003 | −0.016 | −0.064 |
| % mobile home units | −0.301 ** | −0.314 ** | −0.263 ** | −0.309 ** | −0.161 | −0.038 | 0.053 | −0.002 | −0.011 |
| % hous. built after 2000 | 0.069 | 0.063 | −0.013 | 0.051 | 0.356 ** | 0.040 | −0.043 | −0.014 | −0.017 |
| Health prov. per cap. | 0.078 | 0.114 | 0.105 | 0.110 | −0.145 | −0.017 | 0.018 | −0.082 | −0.057 |
| % native people | −0.319 ** | −0.334 ** | −0.297 ** | −0.332 ** | −0.413 ** | −0.055 | 0.051 | 0.050 | 0.024 |
| Employed rate | 0.199 * | 0.188 * | 0.131 | 0.182 * | 0.596 ** | −0.019 | 0.017 | −0.052 | −0.056 |
| % bachelor's degree | 0.314 ** | 0.366 ** | 0.256 ** | 0.347 ** | 0.397 ** | 0.017 | 0.046 | 0.016 | 0.040 |
| Unemployment rate | −0.042 | −0.059 | −0.008 | −0.049 | −0.480 ** | −0.062 | 0.127 | 0.017 | 0.035 |
| Median age | −0.188 * | −0.233 ** | −0.118 | −0.211 * | −0.029 | 0.002 | 0.054 | 0.026 | 0.043 |
| Average hh size | −0.158 | −0.156 | −0.129 | −0.154 | 0.001 | 0.033 | 0.062 | 0.040 | 0.077 |
| Median rent | 0.356 ** | 0.356 ** | 0.310 ** | 0.354 ** | 0.548 ** | 0.011 | −0.077 | 0.051 | 0.012 |
| % ag. employment | −0.199 * | −0.204 * | −0.143 | −0.196 * | −0.318 ** | −0.071 | 0.030 | 0.036 | −0.001 |
| % disability | 0.024 | 0.038 | 0.039 | 0.037 | −0.047 | 0.053 | −0.181 * | 0.090 | 0.047 |
| Mean elev (meters) | −0.138 | −0.196 * | −0.163 | −0.187 * | −0.273 ** | 0.071 | 0.109 | −0.048 | 0.038 |
| % FEMA flood zone | −0.058 | −0.030 | −0.078 | −0.042 | −0.015 | −0.047 | 0.069 | 0.017 | 0.016 |
| % civilian labor force | 0.042 | 0.080 | −0.003 | 0.062 | 0.284 ** | −0.007 | 0.025 | −0.098 | −0.050 |
| % people voted | −0.185 * | −0.176 * | −0.113 | −0.170* | −0.209 * | −0.052 | 0.079 | −0.150 | −0.108 |
| % water area | 0.325 ** | 0.312 ** | 0.332 ** | 0.323 ** | 0.411 ** | −0.045 | −0.119 | −0.045 | −0.100 |
| % fed. exp., disab/retir. | −0.156 | −0.202 * | −0.201 * | −0.200 * | 0.144 | −0.021 | 0.046 | −0.019 | −0.012 |
| % impervious area | 0.631 ** | 0.680 ** | 0.704 ** | 0.691 ** | 0.554 ** | −0.014 | −0.020 | 0.007 | −0.011 |

\*\* Correlation was significant at the 0.01 level (2-tailed). \* Correlation was significant at the 0.05 level (2-tailed).

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
