# Peer review of "Twitter Use in Hurricane Isaac and Its Implications for Disaster Resilience"

_ijgi, doi:10.3390/ijgi10030116_

Round 1

Reviewer 1 Report

The paper deals with quite old data - Twitter use measurement is more than eight years old. However, the implications are quite clear and authors should consider more justify how this old data could be used in current disaster risk management. The conclusions and implications are based on research, where no significant correlations were found between sentiment scores and resilience indicators. This definitely needs further research, which is also suggested by the authors.

I would appreciate more citations and sources related to disaster resilience research. In this case, a higher share of self-citations are presented in the article and should be avoided (especially when there are numbers of other sources which could be cited as well).

In general, the text is very well written, with solid discussion part and also the conclusions are clear and represent the outputs from the article. I would only suggest having better maps - the cartographic visualization presented in the manuscript is unclear, with poor graphic and in some cases, there are interval overlaps (should be fixed).

Author Response

Thank you very much for the comments and suggestions! For details, Please see the attachment.

Reviewer 2 Report

The proposed contribution is an investigation of potential of social media data (in this case Twitter data) in disaster resilience. The authors used a hybrid methodology within the hypothesis of a link between sentiment and message density and the real resilience of territories, including the identification of resilience indices using the RIM model, analysis of Twitter statistics (geospatial message density) and construction of sentiment indices, and their correlation analysis with resilience indices. The association of resilience with message density was confirmed, but not with sentiment indices. The work is satisfactorily explained ed and motivated, and the methodology and findings of the study are discussed in detail and with satisfactory quality.
It is worth noting that the high grades cannot be extended to the illustrations of the articles. The quality of the cartographic material should be improved - currently it is too disharmonious with the quality of the article itself, especially Fig. 1. Counties borders are disappeared, label "Mexico" is placed on the USA partly, etc. 

The following minor inaccuracies in the text of the article have also been pointed out:

21 - "phases" instead of "phrases" 

116, 134, 447 - extra space

The article can be published after revision.

Author Response

(The authors gave the same response as above.)

Reviewer 3 Report

This is, in  general, a fine paper. It could be a little more concise and a good proof reader would remove a number of repetitions. The only item that I would definitely want to see revised is the use of the abbreviations TD and SC in Table 2. Only in the discussion of Table 5, several pages later, did I learn what those terms mean. Please move that note to the discussion of Table 2.

I also wondered why the authors write a paper in 2021 about an event in 2012. So much has changed since in the world of social media. The most obvious explanation is that the rules of Twitter have changed after 2012 and georeferenced data is now much sparser (to a degree that might render similar research in rural areas impossible). This would be worth noting somewhere in the article, best in the Discussion section.

Author Response

(The authors gave the same response as above.)
